# Transitive inference in cleaner wrasses (*Labroides dimidiatus*)

**Takashi Hotta**[1,2,3]*, **Kentaro Ueno**[1], **Yuya Hataji**[2], **Hika Kuroshima**[2], **Kazuo Fujita**[2], **Masanori Kohda**[1]

**1** Department of Biology and Geosciences, Graduate School of Science, Osaka City University, Osaka, Japan, **2** Department of Psychology, Graduate School of Letters, Kyoto University, Kyoto, Japan, **3** Japan Society for the Promotion of Science, Tokyo, Japan

* takasi712000@yahoo.co.jp

## Abstract

Transitive inference (TI) is the ability to infer unknown relationships from previous information. To test TI in non-human animals, transitive responding has been examined in a TI task where non-adjacent pairs were presented after premise pair training. Some mammals, birds and paper wasps can pass TI tasks. Although previous studies showed that some fish are capable of TI in the social context, it remains unclear whether fish can pass TI task. Here, we conducted a TI task in cleaner wrasses (*Labroides dimidiatus*), which interact with various client fishes and conspecifics. Because they make decisions based on previous direct and indirect interactions in the context of cleaning interactions, we predicted that the ability of TI is beneficial for cleaner fish. Four tested fish were trained with four pairs of visual stimuli in a 5-term series: A-B+, B-C+, C-D+, and D-E+ (plus and minus denote rewards and non-rewards, respectively). After training, a novel pair, BD (BD test), was presented wherein the fish chose D more frequently than B. In contrast, reinforcement history did not predict the choice D. Our results suggest that cleaner fish passed the TI task, similar to mammals and birds. Although the mechanism underlying transitive responding in cleaner fish remains unclear, this work contributes to understanding cognitive abilities in fish.

## Introduction

Transitive inference (TI) refers to the ability to estimate relationships between items that have never been presented together based on previous information [1]. For example, from the premises 'A is smaller than B' and 'B is smaller than C', it follows that 'A is smaller than C' although the premise of A and C has not been presented previously. This ability had been thought to be the hallmark of human deductive reasoning [1]. However, McGonigle and Chalmers [2] tested this ability in squirrel monkeys using non-verbal procedures. During training, subjects were trained with a set of four stimuli simultaneous discrimination: A-B+, B-C+, C-D +, and D-E+, in which plus and minus denote reward and non-reward, respectively. After training, they chose D over B in the non-adjacent pair (i.e. a BD test) although the pair had never been presented during training and both stimuli were equally reinforced and non-

**Data Availability Statement:** All relevant data are within the manuscript and its Supporting Information files.

**Funding:** This study was financially supported by KAKENHI (Nos. 25304017, 26540070 and

26118511 awarded to MK and No. H16J09486 and
18J01293 awarded to TH, No. 16H06301 awarded
to KF).

**Competing interests:** The authors have declared
that no competing interests exist.

reinforced. The authors suggested that the monkeys formed a linear hierarchy
(A<B<C<D<E) based on the training pairs and that this was evidence of TI [2].

Some have supported this interpretation of the preference for D over B in the task because
the task does not allow animals to form transitive relationships [1,3]. In verbal tests, after being
taught that B is smaller than C and C is smaller than D, a transitive hierarchy can be inferred,
that is, B<C<D. In a non-verbal task for non-human animals, what is the result of learning
that 'C is reinforced, B is not reinforced' and 'D is reinforced, C is not reinforced'? 'D is rein-
forced, B is not reinforced' may be incorrect because the relationship is not transitive [4]. In
contrast, Wynne et al. [5] argued that if animals respond transitively even when the relations
are not necessarily transitive, the mental process itself might be transitive. Thus, some refer to
the preference for either stimulus in non-adjacent pairs in a TI task as transitive responding
[4,6].

Another important issue is that transitive responding may be based on simpler mechanisms
such as associative models rather than cognitive processes, mental representation, or deductive
reasoning [1]. Associative models propose that the choice of D is the result of the difference in
the relative reinforcement history of the stimuli [7,8]. For example, subjects who received
more rewards by choosing D rather than B would subsequently develop a preference for D
over B even in the absence of rewards [4,6]. Value transfer theory, the first associative model,
showed that reinforcement produced the ordered relationship A<B<C<D<E because of the
bidirectional transfer of associative value between reinforced and non-reinforced stimuli [9].
Later studies found that transfer was not necessary to produce an ordered hierarchy [7,8].
Some researchers used simulation procedures and others calculated the frequency of reward
for each stimulus (B and D) to test whether reinforcement history can predict the choice of D
in a BD test [4,6].

Since the pioneering study on squirrel monkeys [2], researchers have conducted TI tasks
among chimpanzees [10], rhesus monkeys [11], ring-tailed lemurs, mongoose lemurs [12],
brown lemurs, black lemurs [13], rats [14], pigeons [6,9], hooded crows [4], jackdaws [3], pin-
yon jays, western scrub jays [15], Clark's nutcrackers, azure-winged magpies [16], graylag
geese [17], domestic chicks [18], and paper wasps [19]. It was also demonstrated that pinyon
jays, two African cichlid fish and brook trout used TI to assess dominance rank based on
observed social interactions in a controlled laboratory setting [20–24]; however, there is no
study to test TI task in fish other than social context.

Here, we tested whether cleaner wrasses, *Labroides dimidiatus*, respond transitively in a TI
task. Cleaner wrasses, a coral reef fish, are known as the most common cleaner fish that
remove ectoparasites from the body surface of other fish (called 'clients') [25]. They occupy
small territories (called 'cleaning stations') where they interact with clients and have more
than 2000 interactions per day [25]. In the last two decades, cleaner wrasses have become a
model species to examine fish social cognition [26,27]. In fact, there is evidence that they pos-
sess the capability of colour discrimination, social evaluation, and mirror self-recognition [28–
30]. Thus, testing a TI task in this fish would expand the understanding of fish cognition.

Cleaner wrasses are also known as harem fish and exhibit protogyny, where females change
their sex and become male if they grow to be the largest among their group members [31]. In
coral reefs, cleaner wrasse harems have neighbouring harems and females frequently visit
these harems to assess social conditions such as harem size and the body size of members [32].
Females change harem when they find that a neighbouring harem contains fewer larger
females and can change sex and monopolize reproduction earlier [32]. When females assess
neighbouring harems, TI would help determine the dominance hierarchy by observing possi-
ble dyadic interactions among group members; that is, if A is dominant to B and B is dominant
to C, then A would be probably also be dominant to C despite never having observed

interactions between A and C [20,21,24]. Many researchers have suggested that TI has evolved in animals living in large and stable groups with a linear dominance hierarchy (also known as the social complexity hypothesis) [12,13,15,16]. Thus, we predicted that cleaner wrasses have the ability of TI and can pass a TI task [1,5].

For this study, we referred to the procedures used for chimpanzees and rats [10,14]. We trained cleaner wrasses with a potentially hierarchical sequence of five differently coloured stimuli (A-B+, B-C+, C-D+, and D-E+) and subsequently presented a non-adjacent BD pair (i.e. a BD test). To exclude the possibility that training history rendered the associative strength of B greater than that of D, we calculated the reward/non-reward ratio [3,6]. If cleaner wrasses exhibit transitive responding, we would expect that, during the BD test, our subjects would prefer to choose B over D despite training reinforcement history predicting a preference for D over B [1]. We also used two associative models to predict choices in the BD pair based on training history [7,8].

## Materials and methods

### Subjects and acclimation to the experimental set-up

The experiments were conducted in our Laboratory at Osaka City University and Kyoto University using cleaner wrasses, *Labroides dimidiatus*, obtained from commercial breeders. We used four adult females [total length (TL) of 5–6 cm]. The experimental tanks measured $90 \times 30 \times 30$ cm$^3$ and were separated by two partitions (one opaque and one transparent) with a central sliding door ($5 \times 10$ cm) into three compartments [living area ($40 \times 30 \times 30$ cm$^3$), observation area ($10 \times 30 \times 30$ cm$^3$) and experimental area ($40 \times 30 \times 30$ cm$^3$); see Fig 1], and each subject was kept individually in the living area before starting the experiment. The fish were provided with a PVC tube (2.5 cm diameter, 10 cm length) that served as shelter in the living area. The tanks were maintained at $26 \pm 2$ ºC with a 12:12 h light:dark cycle and fed once per day (Tetramin).

The fish were trained to feed on mashed prawn smeared on coloured Plexiglass plates ($6 \times 6$ cm$^2$: red, blue, yellow, green, white, or black) prior to the experiments. One plate was left in the experimental area and both sliding doors were opened. Thus, the subjects had to enter the experimental area to access the plate by passing through the observing area. When the fish ate the food on the plate, we guided them to the living area using a hand net gently. Subsequently, the other plate, which had been refilled with mashed prawn, was presented. During this acclimation phase, all of the coloured plates were presented in random order. Each fish was habituated to enter the experimental area soon after opening the door over five consecutive days. No signs of disease were observed during the study and nor did we observe any stress-related behavior. All subjects were kept in laboratory for other experiments.

### Training stage 1: Acquisition

After acclimation with the experimental setting and plates, the fish were trained with 6 plate pairs of visual stimuli in a 5-term series. We chose five colour plates and assigned one to each stimulus (A, B, C, D, or E); the black and white plates were assigned as B or D to avoid colour bias in the BD test [11]. The colour combination for each subject is shown in S1 Table.

Training stage 1 was separated into six different phases. In the first phase, the subjects were exposed to the first pair (A-B+). A vertical opaque Plexiglass partition was inserted between the two plates to ensure that the fish could access only a single plate and avoid side bias (Fig 1) [33]. Both plates had equal amounts of food on the front side. When the plates were set, the opaque door of the living area was opened and the subjects were allowed to enter the observation area for 10 s. We defined 'choice' as when the tip of the fish's snout first passed the

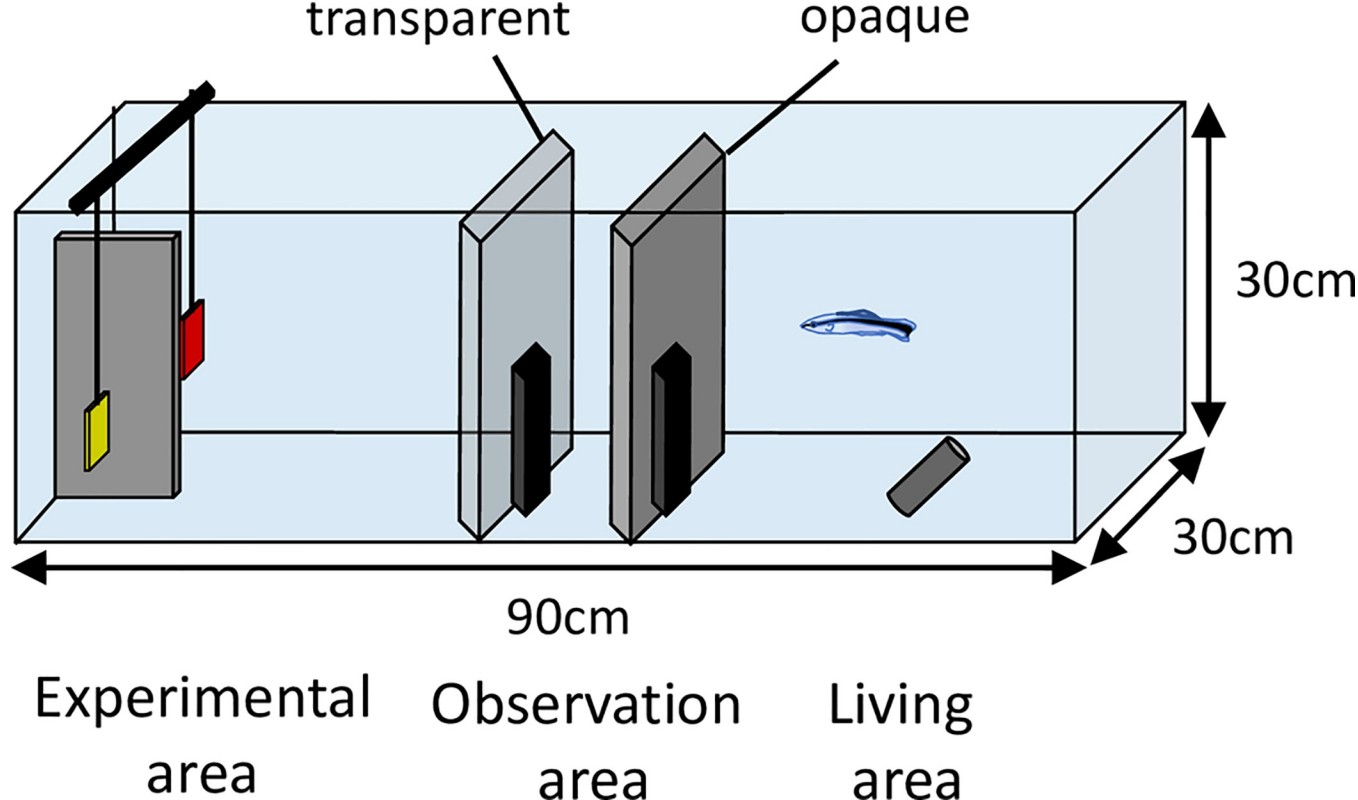

**Fig 1. Schematic drawing of the experimental apparatus.** The tank was separated by opaque and transparent partitions into three compartments (living, observation, and experimental areas). During the experiment, two plates were presented in the experimental area and separated by an opaque partition (see Methods).

threshold of the Plexiglass partition between plates [33]. If subjects chose plate B, they would receive the reward and plate A was subsequently removed. However, if plate A was incorrectly chosen, both plates were removed and the fish could not access the reward. After the fish ate food from a correct plate or explored an incorrect plate for 5 s, they were gently directed back to the living area with the handle of a hand net [34]. The next trial began after an inter-trial interval of 1 min (correct choice) or 2 min (incorrect choice). One session consisted of six trials. The fish participated in up to three sessions per day and the inter-session interval was at least 2 hours. The location of the plates was constrained such that the same plate was never presented more than two consecutive times on the same side. When a fish solved 5/6 or 6/6 trials in two consecutive sessions (binomial test, $p < 0.05$), it was considered to have learned the pair. After reaching this criterion, the fish was transferred to the next phase (B-C+). The sequence of training phase 1 was A-B+, B-C+, A-C+, C-D+, D-E+, and C-E+. Furthermore, we included exposure to two non-adjacent pairs (i.e. A-C+ and C-E+) during training phase 1. This addition might help the subject to integrate stimuli along a common dimension [14].

### Training stage 2: Re-learning

On the subsequent day, after completing phase 6 of training stage 1, to sophisticate learning for adjacent plate pairs, the fish repeated the same training phase except with two differences. First, we eliminated the two non-adjacent plate pairs used in training stage 1 (i.e. A-C+ and C-E+). Second, because training stage 2 immediately followed training stage 1 and incorrect

plates would therefore be seldom chosen, the learning criterion was changed to 5/6 or 6/6 instances of the fish choosing the correct stimulus within one session. When the fish fulfilled the criterion for D-E+ pair, they were transferred to training stage 3 on the next day.

### Training stage 3: Mixed pair exposure

To examine whether the fish learned all adjacent plate pairs, mixed pair exposure was employed within one session. Each plate pair (i.e. A-B+, B-C+, C-D+, and D-E+) was included in two trials within one session in the following order: A-B+, B-C+, C-D+, and D-E+. The learning criteria were: the fish (i) chose over 1/2 in each pair within a session and (ii) solved over 6/8 trials in two consecutive sessions. The inter-session interval was at least 3 hours.

### Test phase

The test phase began on the following day after fulfilling the criteria of training stage 3. In the morning, we conducted mixed pair exposure as in training stage 3 to examine the motivation of the fish. When the fish fulfilled the criteria in training stage 3, the test session was conducted after an interval of 3 hours. In the test session, we presented a non-adjacent plate pair (BD) four times. Because our preliminary study found that the fish did not enter the experimental area when they chose the correct plate but did not receive the reward, we used the same procedure as that in the training phase, that is, the fish could access the food when they chose plate D but not B. The inter-trial interval was also the same as that in the training phase. If the fish fulfilled the learning criterion in the morning, we conducted mixed pair exposure in the afternoon and did not test on that day. This procedure was repeated until each subject fish was tested in three test sessions (i.e. 12 BD test trials).

### Simulations

We simulated the choice in the test phase based on the acquisition of an associative representation using two configural models: one based on the Rescorla-Wagner equations [5] and the other on the Luce equations [7]. Both models were used for pigeons and hooded crows and provided a satisfactory fit for training pairs [4,6]. Lazareva et al. [4] modified these models to incorporate value transfer mechanisms. Thus, we used the Siemann-Delius and Wynne models with or without value transfer modifications. The data of each fish were fitted individually using the full sequence of trials presented during the training phase and employing the least-square error technique. The obtained associative values of the stimuli were used to calculate choice probability for the training pairs (i.e. the AB, BD, CD, and DE pairs) and the BD pair during the test phase according to the choice functions used by the models [4,6]. All simulations were performed by MATLAB and the script described in (S1 File).

### Data analyses

We predicted that cleaner fish would choose D over B after they learned four colour combinations (i.e. A-B+, B-C+, C-D+, and D-E+). To test this prediction, we used the binomial test to test for deviations from chance level in the BD test at the individual level. For the mixed pair exposure prior to the BD test, repeated measures one-way ANOVA was used to examine the difference in the total number of correct choices of each adjacent pair. To evaluate the relative associative strength of the stimuli B and D, we also calculated the reward/non-reward ratio ($R$) by dividing the number of rewards ($N_r$) by the number of non-rewards ($N_n$) in all of the training stages [3]. We used the chi-square test to compare $R_B$ and $R_D$ for each subject. All data

analyses were conducted using R version 3.5.1 (The R Foundation for Statistical Computing, Vienna, Austria; http://www.r-project.org).

### Ethics information

All experiments adhered to the Association for the Study of Animal Behavior's Guidelines for the Use of Animals in Research and were conducted in compliance with the Regulations on Animal Experiments of Osaka City University, Kyoto University, and the Japan Ethological Society. No permits from the Japanese government were needed and our experiments were approved by Kyoto University (No. 19_46). No signs of disease were observed during the study and nor did we observe any stress-related behaviors.

## Results

All four fish learnt all of the training plate pairs over the three training stages and were transferred to the test phase (S1 Fig). For the training premise of plate pairs with mixed pair exposure, the total number of correct trials did not differ among the training pairs (ANOVA, $F(3, 9) = 0.60$, $p = 0.63$). When the non-adjacent BD plate pairs were presented, all subjects chose plate D significantly more than they did plate B (binomial test, $p < 0.05$, S2 Table). The choice of D did not seem to improve during the three sessions (Fig 2) and three fish chose plate D on the first BD test trial (Fish 1, 3, and 4). Finally, we calculated the reward/non-reward ratio ($R = Nr/Nn$) for plates B and D (Table 1). For only one fish (Fish 1), $R_D$ was larger than $R_B$ (chi-square test, $\chi^2(1) = 4.90$, $p < 0.05$), although $R_B$ was larger than $R_D$ for Fish 2 ($\chi^2(1) = 8.00$, $p < 0.05$) and there was an almost significant trend in Fish 3 ($\chi^2(1) = 3.80$, $p = 0.051$). There was no significant difference in Fish 4 ($\chi^2(1) = 0.47$, $p = 0.50$).

We used the Siemann-Delius and Wynne models with or without value transfer modifications to calculate choice probability for the training pairs (i.e. the AB, BD, CD, and DE pairs) and the BD pair during the test phase. The results of the simulations are presented in Fig 3. The individual fits are shown in S4 and S5 Tables, and S3 Table provides model parameter values for the best fitting results. Fig 3 shows that the original and modified models provided a good fit for training pairs. However, the models predicted the average preference for the BD pair to be at or below chance level (S4 and S5 Tables) except in two cases (the Siemann-Delius model with a value transfer mechanism for Fish 1 and the Wynne model for Fish 3).

## Discussion

In this study, we tested whether cleaner fish, *Labroides dimidiatus*, exhibited transitive responding in a TI task. We trained fish to learn the premise plate pairs A-B+, B-C+, C-D+, and D-E+. Subsequently, a non-adjacent BD plate pair was presented. Our results showed that the four tested cleaner fish chose plate D over B, both of which had never been presented together prior to the test phase. This suggests that cleaner fish can respond transitively and is the first demonstration of a TI task in fish [1].

For the BD test, the procedure was the same as that in the training phase in order to maintain motivation; that is, when the fish chose the correct plate, they received a reward. Thus, it is possible that the fish solved the BD test via a rapid learning mechanism [3]. Cleaner fish could discriminate between plates with two different colour patterns within 20 or 30 trials [28]. If so, in the present study, the choice of D in the BD test would improve during the test phase and the BD pair would be learnt as a novel pair (i.e. the AB pair during training phase 1). In contrast with these predictions, we did not find such improvement and the choice of the BD pair (11.00 ± 1.15 trials) tended to be higher than that for novel pairs (6.50 ± 1.91 trials within 12 trials). To exclude the possibility of learning, we also focused on first trial on BD test.

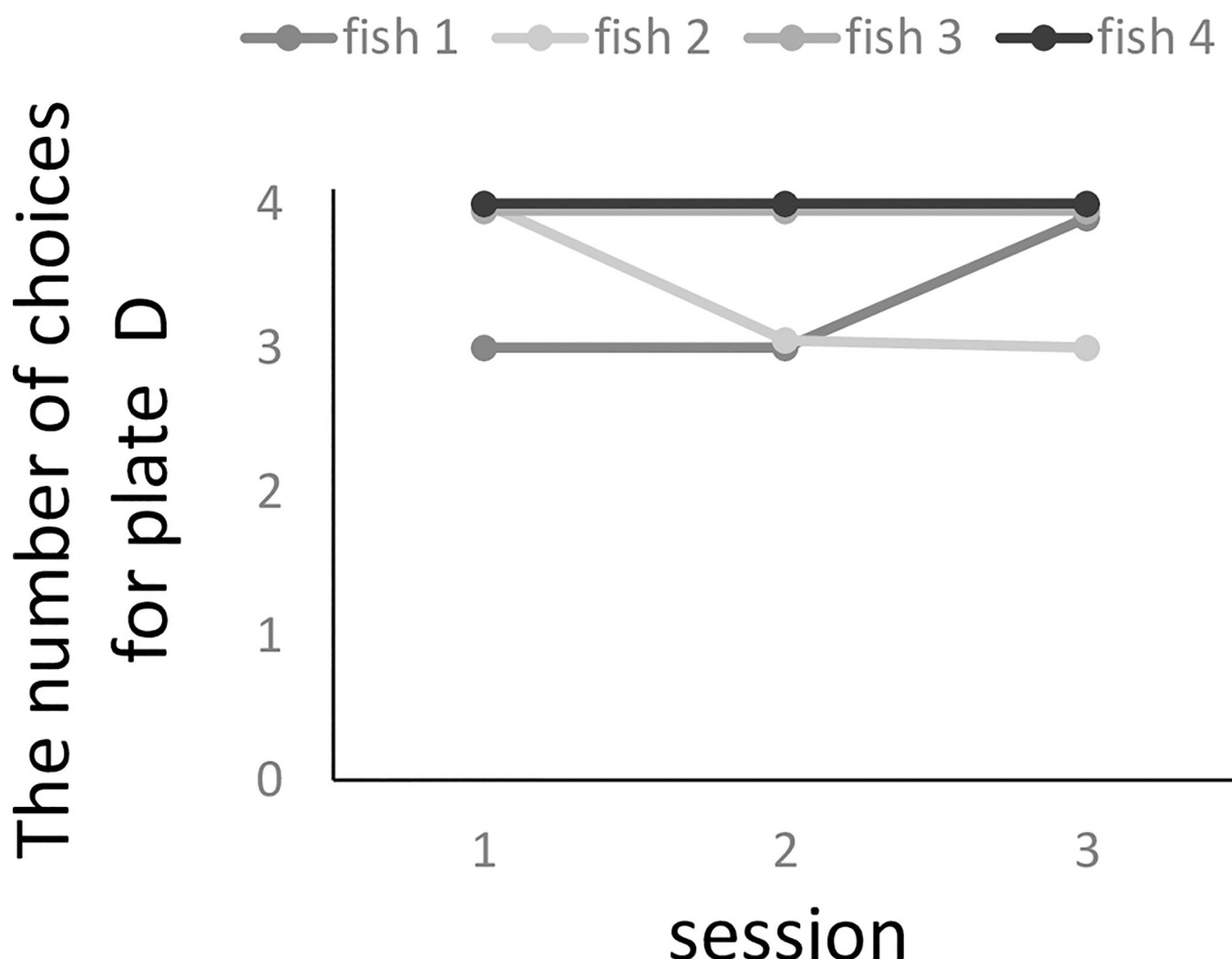

**Fig 2. The number of choices of plate D in the BD test across BD test sessions.** Each session consists of four trials.

If cleaner fish learned the choice of D over B within a few trials, their choices on first trial would be random. However, three fish (fish 1, 3 and 4) chose the D on the first trial on BD test. These indicate that the influence of rapid learning during the BD test was small.

Because our training proceeded in a backward direction (i.e. A-B+, B-C+, C-D+, and D-E +) [1], the possibility remains that the selection of D over B in the BD test trials may reflect a recency effect [15]. That is, our subjects received a reward by choosing D (C-D+) recently on the morning of the test phase. However, it should be unlikely because they were also exposed to choosing D (D-E+) as a non-reward more recently. Additionally, some tested TI task in a backward and forward direction (forward direction; training proceeded in A+B-, B+C-, C+D-, and D+E-) and found that the training direction did not affect the score of BD test [8]. An experiment examining biological market theory found a decreased recency effect in cleaner wrasses [35]. Two plates (resident and visitor plates) were presented for the cleaner fish. When they chose the resident plate, the visitor plate could not be accessed. However, when they chose the visitor plate, they could access the resident plate and receive a reward. If the plate selection of cleaner fish depended on the most recent exposure, they should have chosen the

**Table 1. The number of correct (*Nr*) and incorrect (*Nn*) choices and reward/non-reward ratio (*R*) in training stages 1, 2, and 3.**

|  |  | Fish 1 | Fish 2 | Fish 3 | Fish 4 |
|---|---|---|---|---|---|
| *Nr* |  |  |  |  |  |
|  | B | 38 | 38 | 36 | 46 |
|  | C | 44 | 74 | 58 | 47 |
|  | D | 43 | 55 | 41 | 59 |
|  | E | 95 | 54 | 37 | 19 |
| *Nn* |  |  |  |  |  |
|  | A | 22 | 24 | 14 | 10 |
|  | B | 23 | 44 | 21 | 27 |
|  | C | 33 | 31 | 17 | 11 |
|  | D | 57 | 28 | 9 | 45 |
| *R* |  |  |  |  |  |
|  | B | 1.65 | 0.86 | 1.71 | 4.60 |
|  | D | 0.75 | 1.96 | 4.56 | 5.36 |

resident plate because the trials ended by accessing the resident plate regardless of the plate that was chosen. The results showed that almost all of the cleaners chose the visitor plate initially [35], suggesting that the plate selection of cleaner fish may not depend on their most recent experience.

Although our subjects chose plate D in the BD test, the mechanism underlying the transitive responding remains unclear. Some have argued that transitive responding in a TI task is based on reinforcement history during the training phase (associative models) [1]. To examine this possibility, we calculated the reward/non-reward ratio as the index of the relative associative strength of the stimuli B and D [4,6]. For three fish, $R_D$ was larger than $R_B$, that is, they received the reward more frequently by choosing D than B. Thus, it is possible that D was chosen because of its high relative associative strength. In contrast, for Fish 1, $R_D$ was not larger than $R_B$, suggesting that for at least this fish, relative associative strength based on training history might not explain the choice of D over B. To examine the effect of training history, we used two configural models, the Siemann-Delius and Wynne models, to calculate the possibility of choosing the training pairs and the BD pair during the test phase [4,6]. Both models provided a good fit for the training pairs; however, the probability of choosing D in the BD test was at or below chance level. Taken together, reinforcement history is insufficient to predict the transitive responding in the tested fish, although the mechanism remains unclear.

Grosenick et al. [21] are the first to examine the ability of TI in fish in the social context, using an African cichlid fish, *Astatotilapia burtoni*. In the study, bystander fish observe aggressive interactions between five size-matched conspecific males (A, B, C, D and E). During training, the following contests were presented to the bystander (A+B-, B+C-, C+D- and D+E-, where a plus means a win and a minus denotes a lose). When B and D were presented simultaneously, bystanders avoid fish D, suggesting that they can infer the linear dominance hierarchy (i.e. A>B>C>D>E). Others also tested in another African cichlid fish, *Julidochromis transcriptus*, and Brook trout, *Salvelinus fontinalis*, by using similar procedure [22–24]. These studies revealed TI ability in fish but also conducted with the implicit limitation that TI in fish might be a specialized form of cognition to social context. On the other hand, we demonstrated in a more abstract general procedure developed for mammals and birds [1]. In other words, our study shows that TI ability in fish reported so far are not merely revealing about

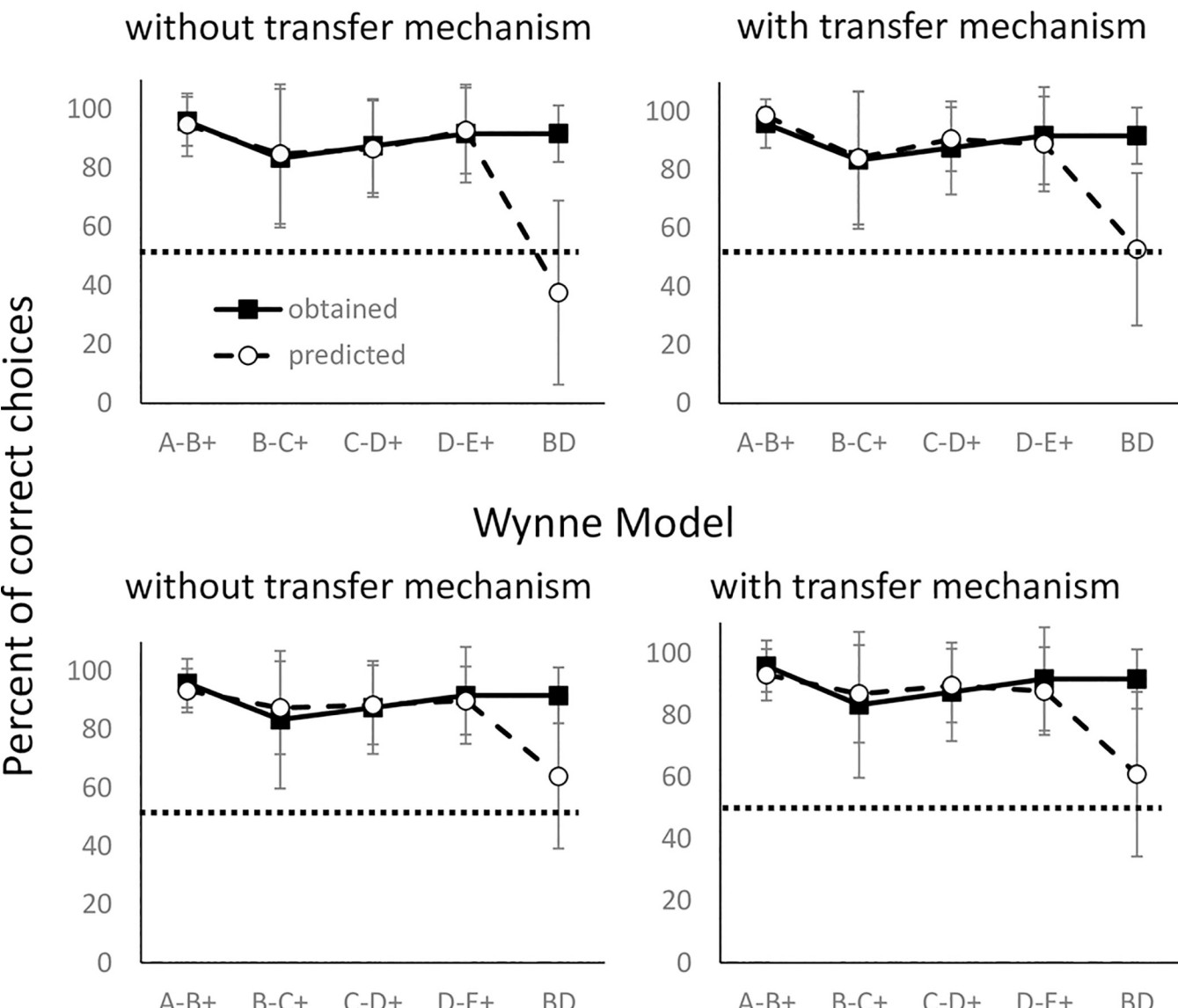

**Fig 3. The obtained percentage of the correct choice in training premise pairs and transitive responses (i.e. choice of plate D) in the tested BD pair compared with simulations using the original and modified Wynne and Siemann-Delius models.** The black squares and white circles denote the obtained and expected percentages, respectively. The dashed lines indicate the chance level (50%).

social cognition, but also fish have a more fundamental inferential capability than previously thought.

It has been assumed that transitive inference evolved in social animals living in large and stable groups [1]. Bond et al. [15] tested this hypothesis to compare performance in a BD test between highly social pinyon jays and less social scrub jays. They found that pinyon jays performed better than did scrub jays. Moreover, the choices of pinyon jays in the BD test were consistent with cognitive accounts but those of scrub jays fitted associative representations [16]. However, social complexity as well as feeding ecology differs between the two species of jays. Pinyon jays have high spatial cognition because they cache many seeds before the winter

season and retrieve them after a long period. If transitive inference is supported by spatial representation [1], their sophisticated spatial cognition may develop the ability of transitive inference [12]. To test this possibility, they added two species of jays that differ in social complexity and/or feeding ecology from pinyon and scrub jays [16]. The results showed that both social and feeding ecology influenced performance in the BD test. In contrast, comparative studies in prosimian primates and lemurs did not support the hypothesis that transitive inference evolved in social animals [12,13]. Cleaner wrasses have complex interactions with conspecifics and heterospecifics, but spatial cognition does not differ from other labroid fishes [36]. Thus, labroid fishes are good candidates to test whether social complexity or spatial cognition develops the ability of transitive inference.

In this study, we revealed that cleaner wrasses can respond transitively in a TI task. To the best of our knowledge, this is the first documentation of transitive responding in fish. The model simulation results did not support the possibility that the choice of plate D in the BD test was predicted by reinforcement history; however, the mechanism underlying the transitive responding remains unclear. Thus, future studies with modified procedures, such as the circular task, should examine whether fish can form mental representations based on training premise pairs [11]. The hippocampus is thought to play a central role in transitive inference [37]. A recent study found a structure homologous to the hippocampus in the fish brain (dorsolateral telencephalon) [27]. Thus, neurobiological investigation would also contribute to revealing the mechanism of transitive responding in fish.

## Supporting information

**S1 Fig. The number of sessions for fulfilling the criteria during the training stages for each subject.**
(PDF)

**S1 Table. The color type used for each subject.**
(PDF)

**S2 Table. The number of correct choices out of total number of adjacent pair trials before BD test and number of choices of D on BD test in test phase.** Asterisk means $p < 0.05$ (binomial test).
(PDF)

**S3 Table. Each parameters and least-square difference of best fitted models (the original and modified Siemann-Delius and Wynne).** LSD means the least-square difference between the proportion of correct responses predicted by the model and obtained in the training phase. ß, ε, α, γ, A are the model parameters [4].
(PDF)

**S4 Table. Simulated accuracies for the Siemann-Delium and Wynne models.**
(PDF)

**S5 Table. Simulated accuracies for the modified Siemann-Delium and Wynne models incorporating a value transfer mechanism.**
(PDF)

**S6 Table. The raw data for individual fish.**
(XLSX)

**S1 File. MATLAB script for simulations.**
(ZIP)

## Acknowledgments

We would like to thank Dr. Tomohiro Takeyama and the members of Laboratory of Animal Sociology, Osaka City University and Laboratory of Psychology, Kyoto University for fruitful discussions regarding this work.

## Author Contributions

**Conceptualization:** Takashi Hotta, Kentaro Ueno, Masanori Kohda.

**Data curation:** Takashi Hotta, Kentaro Ueno.

**Formal analysis:** Takashi Hotta, Kentaro Ueno.

**Funding acquisition:** Takashi Hotta, Masanori Kohda.

**Investigation:** Takashi Hotta, Kentaro Ueno, Yuya Hataji.

**Methodology:** Takashi Hotta, Kentaro Ueno, Hika Kuroshima, Kazuo Fujita, Masanori Kohda.

**Project administration:** Takashi Hotta, Masanori Kohda.

**Resources:** Takashi Hotta, Masanori Kohda.

**Software:** Takashi Hotta, Yuya Hataji.

**Supervision:** Masanori Kohda.

**Validation:** Masanori Kohda.

**Visualization:** Takashi Hotta.

**Writing – original draft:** Takashi Hotta.

**Writing – review & editing:** Takashi Hotta, Hika Kuroshima, Kazuo Fujita, Masanori Kohda.

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
