## [Decision Letter · Decision Letter 0]

8 Jun 2020

PONE-D-20-08282

Transitive inference in cleaner wrasses (Labroides dimidiatus)

PLOS ONE

Dear Dr. Hotta,

Thank you for submitting your manuscript to PLOS ONE. After careful consideration, we feel that it has merit but does not fully meet PLOS ONE’s publication criteria as it currently stands. Therefore, we invite you to submit a revised version of the manuscript that addresses the points raised during the review process.

Although your paper seems to rely on sound methods and results and brings interesting new conclusions to the field, I agree with the reviewer that you should make the scripts avaliable and the progression of the models tested with respect to experimental results. Also pooease add the asked paragraph abour earlier fish research with dominance hierarquies and TI.

We look forward to receiving your revised manuscript.

Kind regards,

Nicolas Chaline

Academic Editor

PLOS ONE

Journal Requirements:

2. Please provide details concerning the final disposition of the animals at the end of the study, and describe any procedures for euthanasia.

Additional Editor Comments (if provided):

Reviewers' comments:

Reviewer's Responses to Questions

**Comments to the Author**

1. Is the manuscript technically sound, and do the data support the conclusions?

Reviewer #1: Yes

2. Has the statistical analysis been performed appropriately and rigorously? 

Reviewer #1: Yes

3. Have the authors made all data underlying the findings in their manuscript fully available?

Reviewer #1: Yes

4. Is the manuscript presented in an intelligible fashion and written in standard English?

Reviewer #1: Yes

5. Review Comments to the Author

Reviewer #1: In their manuscript, "Transitive inference in cleaner wrasses (Labroides dimidiatus)" authors Hotta, Ueno, Hataj, Kuroshima, Fujita, and Kohda describe the results of a transitive inference procedure conducted using cleaner wrasses. They assert (correctly, so far as I am aware) that this is the first demonstration of TI in fish that does not depend on a comparison of social rank.

For the most part, the experiment that the authors have conducted what in the abstract appears to be a very standard training procedure for a TI experiment. Their main innovation is to describe how this can be done in fish using stimuli that are more abstract in character than has been demonstrated previously, and they have included a sufficient level of detail that an interested party could replicate the experiment without difficulty. This on its own is a substantial contribution, as it provides a template for implementing this cognitive probe in a much wider range of species. The analysis and interpretation of the data are clear and convincing. On its experimental merits alone, I think this manuscript is suitable for publication.

My main advice for improving the manuscript relates to the theoretical calculation of associative strength using value transfer models. The reader is told that these models were fit "with or without value transfer modifications" but any further details about this implementation are omitted. Since the analyses were conducted in R, I think it would be beneficial to future work in this area for the authors to add an annotated R script to their supplemental information that a rader could download and run to replicate the process of fitting parameters using these models, in order to confirm the predicted behavior at test that is reported in Figure 3. Since the comparison of behavior to that of associative models is central to the claims made in the paper, the authors would do the field a favor by showing their work at this stage.

I also think the authors should devote at least a paragraph of their discussion to comparing this study to other studies that have examined TI in fish. The work by Grosenick and colleagues (2007) was surprising to at least a few comparative psychologists. That study's reliance on dominance hierarchies set the stage for subsequent TI procedures, but also carried with it the implied limitation that this might be a specialized form of cognition, limited to the social arena. The authors should revisit more explicitly the claim they make in the introduction that the current study demonstrates TI in a more abstract way than has been accomplished in fish to date, which in turn should strengthen the reader's confidence that past reports of TI in fish are not merely quirks of social cognition, but may indeed reflect a more fundamental inferential capacity.

6. PLOS authors have the option to publish the peer review history of their article (what does this mean?). If published, this will include your full peer review and any attached files.

Reviewer #1: Yes: Greg Jensen

---

## [Author Response · Author response to Decision Letter 0]

11 Jun 2020

Comment 1

In their manuscript, "Transitive inference in cleaner wrasses (Labroides dimidiatus)" authors Hotta, Ueno, Hataji, Kuroshima, Fujita, and Kohda describe the results of a transitive inference procedure conducted using cleaner wrasses. They assert (correctly, so far as I am aware) that this is the first demonstration of TI in fish that does not depend on a comparison of social rank.

For the most part, the experiment that the authors have conducted what in the abstract appears to be a very standard training procedure for a TI experiment. Their main innovation is to describe how this can be done in fish using stimuli that are more abstract in character than has been demonstrated previously, and they have included a sufficient level of detail that an interested party could replicate the experiment without difficulty. This on its own is a substantial contribution, as it provides a template for implementing this cognitive probe in a much wider range of species. The analysis and interpretation of the data are clear and convincing. On its experimental merits alone, I think this manuscript is suitable for publication.

Response 1

We wish to express our appreciation to Reviewer #1 for his (or her) insightful comments, which have helped us significantly improve the paper. As pointed out, we added the information on simulations and the discussion the difference in previous study in other fish.

Comment 2

My main advice for improving the manuscript relates to the theoretical calculation of associative strength using value transfer models. The reader is told that these models were fit "with or without value transfer modifications" but any further details about this implementation are omitted. Since the analyses were conducted in R, I think it would be beneficial to future work in this area for the authors to add an annotated R script to their supplemental information that a reader could download and run to replicate the process of fitting parameters using these models, in order to confirm the predicted behavior at test that is reported in Figure 3. Since the comparison of behavior to that of associative models is central to the claims made in the paper, the authors would do the field a favor by showing their work at this stage.

Response 2

Thank you for your advices. All simulations were performed by MATLAB and we presented the script as supplemental file (Zip file). We added this sentence in L. 233-234.

Comment 3

I also think the authors should devote at least a paragraph of their discussion to comparing this study to other studies that have examined TI in fish. The work by Grosenick and colleagues (2007) was surprising to at least a few comparative psychologists. That study's reliance on dominance hierarchies set the stage for subsequent TI procedures, but also carried with it the implied limitation that this might be a specialized form of cognition, limited to the social arena. The authors should revisit more explicitly the claim they make in the introduction that the current study demonstrates TI in a more abstract way than has been accomplished in fish to date, which in turn should strengthen the reader's confidence that past reports of TI in fish are not merely quirks of social cognition, but may indeed reflect a more fundamental inferential capacity.

Response 3

Thank you for your suggestion. We agree that our study strengthen previous researches on transitive inference in fish. Previous studies (e.g. Grosenick et al. 2007) surely revealed that some fish have a capable of TI in social context, but it conducted with the implicated limitation that it may be a specialized form of cognition in dominance relationship. However, we demonstrated in a similar way such as mammals and birds in more abstract context. Thus, our results indicated that the TI ability found in previous studies are not merely social cognition, but also reflect a more fundamental inference capability in fish.

Like this, we added a paragraph for discussion in L. 344-357.

---

## [Decision Letter · Decision Letter 1]

4 Aug 2020

Transitive inference in cleaner wrasses (Labroides dimidiatus)

PONE-D-20-08282R1

Dear Dr. Hotta,

We’re pleased to inform you that your manuscript has been judged scientifically suitable for publication and will be formally accepted for publication once it meets all outstanding technical requirements.

Kind regards,

Nicolas Chaline

Academic Editor

PLOS ONE

Additional Editor Comments (optional):

Reviewers' comments:

Reviewer's Responses to Questions

**Comments to the Author**

1. If the authors have adequately addressed your comments raised in a previous round of review and you feel that this manuscript is now acceptable for publication, you may indicate that here to bypass the “Comments to the Author” section, enter your conflict of interest statement in the “Confidential to Editor” section, and submit your "Accept" recommendation.

Reviewer #1: All comments have been addressed

2. Is the manuscript technically sound, and do the data support the conclusions?

Reviewer #1: Yes

3. Has the statistical analysis been performed appropriately and rigorously? 

Reviewer #1: Yes

4. Have the authors made all data underlying the findings in their manuscript fully available?

Reviewer #1: Yes

5. Is the manuscript presented in an intelligible fashion and written in standard English?

Reviewer #1: Yes

6. Review Comments to the Author

Reviewer #1: In their manuscript, "Transitive inference in cleaner wrasses (Labroides dimidiatus)" authors Hotta, Ueno, Hataj, Kuroshima, Fujita, and Kohda describe the results of a transitive inference procedure conducted using cleaner wrasses.

As noted in my previous review, I already considered the manuscript to be in satisfactory condition to warrant acceptance, and I remain enthusiastic about its publication. I would like to thank the authors to taking my additional small comments on board, particularly the inclusion of their simulation scripts. I look forward to seeing it in digital print, so to speak.

7. PLOS authors have the option to publish the peer review history of their article (what does this mean?). If published, this will include your full peer review and any attached files.

Reviewer #1: **Yes: **Greg Jensen

---

## [Editor Report · Acceptance letter]

7 Aug 2020

PONE-D-20-08282R1 

Transitive inference in cleaner wrasses (*Labroides dimidiatus*) 

Dear Dr. Hotta:

I'm pleased to inform you that your manuscript has been deemed suitable for publication in PLOS ONE. Congratulations! Your manuscript is now with our production department. 

Kind regards, 

on behalf of

Professor Nicolas Chaline 

Academic Editor

PLOS ONE